# Physiological Properties of Perennial Rice Regenerating Cultivation in Two Years with Four Harvests

**DOI:** 10.3390/plants12223910

**Published:** 2023-11-20

**Authors:** Chunlin Guo, Weiwei Lin, Wujie Gao, Chaojie Lan, Hailong Xu, Jingnan Zou, Nyumah Fallah, Wenfei Wang, Wenfang Lin, Ting Chen, Wenxiong Lin

**Affiliations:** 1Fujian Key Laboratory for Crop Physiology and Molecular Ecology, College of Agriculture, Fujian Agriculture and Forestry University, Fuzhou 350002, China; guochunlin0516@163.com (C.G.); elianagao0222@163.com (W.G.); zoujingnan222@163.com (J.Z.);; 2Fujian Key Laboratory for Agroecological Processes and Safety Monitoring, College of Ecology, Fujian Agriculture and Forestry University, Fuzhou 350002, China; 3College of Life Sciences, Fujian Agriculture and Forestry University, Fuzhou 350002, China

**Keywords:** perennial rice, regeneration characteristics, endogenous hormones, basal tillering nodes, ratoon crop

## Abstract

Crop perennialization has garnered global attention recently due to its role in sustainable agriculture. However, there is still a lack of detailed information regarding perennial rice’s regenerative characteristics and physiological mechanisms in crop ratooning systems with different rice stubble heights. In addition, the response of phytohormones to varying stubble heights and how this response influences the regenerative characteristics of ratoon rice remains poorly documented. Here, we explored the regenerative characteristics and physiological mechanisms of an annual hybrid rice, AR2640, and a perennial rice, PR25, subjected to different stubble heights (5, 10, and 15 cm). The response of phytohormones to varying stubble heights and how this response influences the regenerative characteristics of ratoon rice were also investigated. The results show that PR25 overwintered successfully and produced the highest yield, especially in the second ratoon season, mainly due to its extended growth duration, higher number of mother stems, tillers at the basal nodes, higher number of effective panicles, and heavier grain weight when subjected to lower stubble heights. Further analysis revealed that PR25 exhibited a higher regeneration rate from the lower-position nodes in the stem with lower stubble heights. this was primarily due to the higher contents of phytohormones, especially auxin (IAA) and gibberellin (GA_3_) at an early stage and abscisic acid (ABA) at a later stage after harvesting of the main crop. Our findings reveal how ratoon rice enhances performance based on different stubble heights, which provides valuable insights and serves as crucial references for delving deeper into cultivating high-yielding perennial rice.

## 1. Introduction

The rapid global population expansion has increased the importance of finding more diverse food and energy sources to support human well-being. The demand for food, including rice, is expected to rise significantly, as the world population is projected to reach 9 billion people by 2050 [1], and due to climate change [2]. These ever-increasing demands have led to the industrialization and modernization of many industries, including agriculture, food processing, etc., which is extremely important for promoting production diversity and increasing the quantity and quality of products [3]. For example, farmers heavily rely on agricultural inputs, including synthetic fertilizers, insecticides, pesticides, etc., to address these challenges [4]. Mosier et al. documented the extensive role that fertilizers, especially nitrogen (N) fertilizer, play in food production [5]. They observed that approximately 40% of humans rely on N fertilizer to produce food. It was also documented that N fertilizer accounts for 56% of rice, maize, and wheat production [6]. Additionally, Li and Yang [7] reported that the amount of N fertilizer (400 kg N·ha^−1^) used in China for perennial crop production, such as sugarcane, exceeds the average N required worldwide. However, indiscriminate application of these agricultural inputs can adversely affect soil fertility, health, fitness, and the overall environment [8]. Carpenter et al. studied the non-point contamination of surface waters induced by N and phosphorus (P) and noticed that eutrophication was a common phenomenon in lakes, rivers, estuaries, and coastal oceans, primarily due to P and N over-enrichment. A promising and unique farming system called the perennial/ratoon rice farming system has come to light as a potential answer to these problems [9]. 

A perennial farming system involves cultivating crops that can be maintained and harvested across several growing seasons and years [10]. This farming system is resilient to flooding, extreme weather, and storms. In this farming system, some commonly cultivated crops include plantain, banana, wheat, sorghum, rice, etc. [11]. Studies have revealed that the axillary buds on perennial rice (PR) nodes have the ability to overwinter and subsequently develop into new plants in the following seasons [12], resulting in lower agricultural inputs. Over recent decades, perennial farming systems, especially rice perennialization, have gained traction among many farmers [12]. For instance, Hu et al. [13] extensively explored the use of perennial African rice *Oryza longistaminata* as a donor and a local elite variety of annual domesticated Asian rice as a receptor using multi-generation backcrossing of the donor line with the receptor parent. These authors successfully developed nine new PR varieties, among which PR Yunda 23 (PR23), PR Yunda 25 (PR25), and PR Yunda 107 (PR107) have been approved and commercialized as perennial crop varieties worldwide. They also documented that PR 107 exhibited significantly higher productivity [14,15,16]. In a related study, these cultivars were well suited to growing in indica rice areas at altitudes lower than 1550 m above sea level in Yunnan province primarily due to their ability to overwinter effectively, the stable yield performance between the first season and the regenerated seasons, moderate growth duration, stronger disease resistance, high fertilizer tolerance, and higher survival rates of axillary buds in the remaining stubble (86–90%) [12]. Li et al. also found that perennial rice regenerated from the remaining rice stubble for 4–5 years with 8–10 harvests and produced a grain yield of about 15 t·hm^−2^ each year [17]. Further studies on the newly developed perennial rice focused on the interaction of the genotype and the environment and provided valuable information on perennial rice varieties’ ecological and genetic adaptability [12,15,18]. However, there is still a lack of detailed information regarding perennial rice’s regenerative characteristics and physiological mechanisms in crop ratooning systems with different stubble heights. Furthermore, the response of plant hormones to varying stubble heights and how this response influences the regenerative characteristics of ratoon rice remain poorly documented. This study explored the regenerative characteristics and physiological mechanisms of an annual hybrid rice, AR2640, and a perennial rice, PR25, subjected to different stubble heights (10, 15, and 20 cm). The response of plant hormones to varying stubble heights and how this response influences the regenerative characteristics of ratoon rice were also investigated. 

## 2. Results

### 2.1. Weather Conditions and Safe Overwintering Evaluation

In this study, the field trial design and the change in growth duration of rice accessions are illustrated in Table 1. Additionally, the study site’s basic information, such as daily meteorological data encompassing the minimum and maximum air temperature, as well as the total precipitation throughout the experimental period, was recorded (Figure 1). 

On 8 March, the lowest recorded temperature was 7.5 °C, and the highest temperature of 41.8 °C was observed on 29 July. Throughout the first and second harvest seasons (March–November), the total rainfall recorded was 1211.2 mm. Notably, during the period from November 2019 to March 2020, we observed that perennial rice (PR25) exhibited successful overwintering compared to annual rice (AR2640). This observation suggests a higher survival rate of axillary buds in the remaining stubble of PR25 after the harvest of PR25-FRR. 

### 2.2. Yield Attributes of Perennial and Annual Rice Accessions

AR2640 and PR25 yields in rice regrowth were investigated at different cut heights of the leftover stubble over four harvest seasons spanning two years (Table 2). AR2640 in the main cropping season (AR2640-MC) produced a significantly higher yield than PR25 in the same season (PR25-MC) when using the rice ratooning system in 2019. Sequentially, in the first regenerating season of PR25-MC and AR2640-MC, the average grain yields of PR25-FRR and AR2640-FRR generated from PR25-MC and AR2640-MC, respectively, significantly decreased by 31.73% and 64.20%, respectively, and their average growth duration decreased by 44.42% and 49.47%, respectively, relative to the two main crops at the three cutting stubble heights. Further analysis displayed that, on average, PR25-FRR produced a higher yield than AR2640-FRR at lower cutting stubble heights of the main crop. Moreover, the highest yield of PR25-FRR was generated at lower cut heights of stubble (5–10 cm), and the growth duration did not increase with the decrease in the cutting stubble height of PR25 (Table 1). However, AR2640-FRR showed the opposite trend, indicating a higher yield and shorter growth duration at the higher cutting stubble height (15 cm) compared to the plants subjected to 10 cm and 5 cm stubble heights. This finding suggests that a lower cutting stubble height was conducive to the high-yielding formation of perennial rice PR25, whereas the reverse was true in the case of annual rice AR2640. 

PR25-FRR was able to successfully overwinter, whereas AR2640-FRR could not survive after being harvested. During the main season from March to August in 2020, PR25-FRR regenerated into the new ratoon crop (i.e., PR25-SRR) from the different stubble cut heights, and the average yield of PR25-SRR was 49.75%, 26.39% higher than those of PR25-FRR and PR25-MC despite the average growth duration of PR25-SRR being reduced by 36.03% and 14.29% compared with that of PR25-FRR and PR25-MC, respectively (Table 1). Additionally, PR25-SRR regrew into a subsequent ratoon crop (PR25-TRR), which demonstrated a significantly decreased average grain yield compared to the earlier ratoon generations, PR25-SRR and PR25-FRR (62.18% and 24.75% decrease, respectively). The growth duration of PR25-TRR was also reduced by 36.06% compared with that of PR25-SRR, but slight differences were found in the growth duration between PR25-TRR and PR25-FRR and those under the three stubble cut height treatments. Further analysis suggested that PR25-SRR exhibited a much higher number of effective panicles on average, explaining why PR25-SRR produced a higher yield than its original main crop. We also found that, relative to the original main crop and the earlier-generation ratoon crops, such as PR25-FRR and PR25-SRR, PR25-TRR had the lowest number of effective panicles and the smallest number of grains per panicle, consequently producing the lowest output in the two-year rice ratooning cycle, as shown in Table 2. 

### 2.3. Relationship between Axillary Bud Sprout and Endogenous Hormones 

#### 2.3.1. Effect of Stubble Height on Regeneration Rate and Effective Panicles 

Stubble height had different effects on the bud regeneration rate and number of effective panicles of both AR2640 and PR25 (Table 3). For AR2640, the regeneration rate of axillary buds and the number of effective panicles increased with increasing cut height of rice stubble. This effect was particularly pronounced when the stubble height was 15 cm compared to 10 cm and 5 cm stubble heights. However, PR25 exhibited the opposite trend. When subjected to the 5 cm and 10 cm stubble heights, PR25 had a significantly higher regenerative rate and number of effective panicles during the first harvest compared to with the 15 cm stubble height. Moreover, PR25 demonstrated the highest regenerative rate and number of effective panicles in main crop and the fouth harvests when subjected to the 5 cm and 10 cm stubble heights. 

Additionally, the effective panicles of AR2640 were prevalent on the first and second basal nodes in the stem of remaining stubble, accounting for 66.6% and 33.4% of the total effective panicles under the 5 cm stubble height regime, respectively. Furthermore, the effective panicles and grain yield were increased with increasing stubble heights, as shown in Figure 2 and Table 2, Table 3 and Table 4. The third and fourth effective nodal panicles of AR2640 contributed 35–40% of the grain yield, suggesting that higher stubble heights resulted in higher yields for AR2640. However, PR25 produced the highest number of effective panicles at the 5 cm stubble height, which were more pronounced on the 1st~3rd nodes from the basal stems of the remaining stubble, accounting for more than 90% of the total panicles, as evident in the pattern shown in Figure 2 and Table 2, Table 3 and Table 4. Moreover, the highest effective panicle rate (>50%) was generated from the first node of the stem base at the lowest stubble height (5 cm). Additionally, it was noted that the total number of effective panicles and the grain yield decreased with increasing stubble height (Table 2, Table 3 and Table 4), whereas AR2640 revealed the opposite trend, implying that a lower stubble height promoted PR25 growth and yield. 

#### 2.3.2. Sprouting and Elongating Dynamics of Axillary Buds 

The growth properties of the crop’s axillary buds at different nodes of the stem on the remaining rice stubble responded to apex dominance. This response is associated with the species and ratios of endogenous phytohormones in the ratooning process of the two rice varieties. The sprout and elongation of the axillary buds at the base of the stems was also detected in the late growth stage of the main crops, AR2640-MC and PR2-MC, under the same cultivation conditions (Table 4, Figure 3). This suggests that, from the booting stage to the maturity stage of the crops, the length of the axillary buds at the third stem base node and internode was larger than that of the first and second base nodes.

It was revealed that AR2640 exhibited higher elongation in the axillary bud than PR25, especially from the booting stage to the heading stage, suggesting that AR2640 had stronger apex dominance in axillary bud sprouting and elongation relative to PR25. This finding further confirmed that the higher stubble height was conducive for AR2640 in rice ratooning, which might have resulted from the apex dominance mediated by the species and crop phytohormone ratios. 

Further analysis revealed that the stubble heights significantly affected the sprouting and elongation dynamics of the regenerated tillers in both rice varieties after harvest, as shown in Figure 3. The axillary bud lengths increased with the increase in the axillary bud node positions. The average axillary bud lengths of the two varieties were measured at 1–3 cm on day (d) 2 after harvesting. Subsequently, the axillary bud lengths extended and exhibited the fastest growth rate from 7 to 12 and from 12 to 22 d after harvest, respectively. By 12 d, the axillary buds were three times longer than 7 d after harvest. Furthermore, the axillary bud lengths of AR2640 and PR25 at the third node position from the base were significantly higher than those at the first node position 22 d under different stubble heights after harvest. Further analysis showed that the basal internodes of PR25 were dense and short. The axillary buds of the three basal nodes were retained without being affected at the 5 cm stubble height, whereas AR2640 showed the opposite trend. 

#### 2.3.3. Changes in Endogenous Hormone Content of Axillary Buds

The variations of phytohormones, including auxin (IAA), abscisic acid (ABA), gibberellin (GA_3_), and strigolactone (SL), in ratooning rice regeneration were investigated. These hormones were detected in axillary bud organs at different time points and stubble heights after harvest.

##### Dynamic Changes in IAA Content

PR25 exhibited a higher average IAA content in the axillary buds than AR2640, especially in low-position node buds from 2 d to 17 d after harvest as a whole (Figure 4). Moreover, the IAA content of both AR2640 and PR25 increased from 2 to 12 d and decreased from 12 d to 22 d, with 12 d accounting for the maximum amount of IAA after harvest. However, no significant difference was observed in axillary buds at the same node position of the same cultivar when comparing different stubble heights. In addition, the content of IAA in axillary buds at the low node position was higher than that at the high node position in both cultivars at different stubble heights. It was also revealed that IAA in PR25 was significantly higher at the low node position compared with the high node position from 2 to 22 d. However, significant differences were not always observed in AR2640 axillary buds at the high and low node positions from 2 to 22 d.

##### Dynamic Changes in GA_3_ Content

As shown in Figure 5, GA_3_ content in AR2640 was significantly higher than that in PR25 at the low- and high-position node buds at different stubble heights, with 17 d accounting for the maximum amount of GA_3_ after harvest on the whole. Additionally, we found that there were no differences in the GA_3_ content in PR25 at the high-position nodes under different stubble heights. However, the GA_3_ content in PR25 was significantly higher at the low-position node buds than that at the high-position node buds under different stubble heights. This finding suggests that PR25’s high regenerative rate at lower stubble heights was associated with higher contents of IAA and GA_3_. We also believe that the high contents of IAA and GA_3_ identified at a high stubble height (15 cm) may explain the high regenerative characteristics of AR2640 (Figure 4 and Figure 5).

##### Dynamic Changes in ABA Content

The ABA content in the axillary buds of AR2640 and PR25 with different stubble heights is shown in Figure 6. Overall, the content of ABA in the axillary buds of AR2640 and PR25 showed a linear increase with time at different stubble heights. This suggests that the changes in ABA content were linked to the senescence of the rice plants. The ABA content in AR2640 axillary buds increased after 7 d in the presence of different stubble heights and subsequently decreased due to the increase in IAA and GA_3_ levels. After 12 d, the content of ABA in the axillary buds of AR2640 exhibited an increasing trend. Moreover, the ABA content in axillary buds at the low stubble height was higher relative to the high stubble height. In addition, PR25’s ABA content in the axillary buds of the crops gradually increased when subjected to the various stubble heights, reaching its highest peak on day 22. 

##### Dynamic Changes in SL Content

In Figure 7, the SL content in the axillary buds of both AR2640 and PR25 was significantly higher at low stubble heights compared to at high stubble heights 2 d after harvest. In addition, the SL content of the axillary buds at each node of AR2640 showed a decreasing trend within 7 d after harvest and subsequently increased. However, the SL content in the axillary buds of PR25 decreased sharply 12 d but increased slightly 22 d after harvest at different stubble heights. These findings further suggest that the changes in the contents of ABA and SL in the axillary buds were associated with the senescence of the rice and the interaction with the other phytohormones, such as IAA and GA_3_. 

##### Correlation Analysis between the Endogenous Hormone Content and Regeneration Rate

Correlation analysis revealed a significant negative correlation between the SL content and the regenerative rate of the crops 2 d after harvest (r = −0.617 *). It was noticed that both GA_3_ and SL were positively correlated with the regenerative rate of the crops after 17 d (r = 0.580 * and r = 0.677 *), respectively. However, no significant correlation was observed between ABA and the regenerative rate of the crop on 7, 12, and 22 d after harvesting main crop (Table 5).

### 2.4. Changing Characteristics of Rice Root Activity 

The results reveal that the root activity of PR25 was significantly higher compared to AR2640 at all growth stages. This difference was particularly prominent during all booting stages of both the AR2640 and PR25 main crops, as well as the ratooning rice crops. The crops’ stubble height did not significantly affect root activity in the ratooning season rice of AR2640 and PR25, indicating that the root activity was associated with the properties of the rice accessions used. It was also noticed that PR25 exhibited stronger root vigor in all growth stages, leading to a higher regenerative ability than AR2640 in rice ratooning (Table 6).

Further correlation analysis showed a significant positive correlation between the rice regenerative rate and root activity at the main crop’s booting stage, full heading stage, and maturity stage (r = 0.952 **, 0.887 **, 0.935 **, respectively). The crops’ regenerative rate was also positively correlated with the root activity from the booting stage to the maturity stage (r = 0.861 *, 0.869 **, 0.971 **, respectively). It was noticed that there was a significant positive correlation (r = 0.921 **) between the crops’ regenerative rate and their grain yield. Moreover, the grain yield of the crops exhibited a significant positive correlation with the root activity of the crop during the booting and the maturity of the main crops (r = 0.857 *, 0739 *, 0.846 *, respectively). The analysis also revealed a similar correlation pattern between the grain yield and the root activity during the booting stage and the ripening stage of the crops (r = 0.700 *, 0.546 *, 0.695 *, respectively) (Table 7).

## 3. Discussion

Rice ratooning has gained traction among rice farmers due to its low production cost and better economic returns [19]. In recent years, Hu and his team have successfully bred a perennial male wild rice (*O. longistaminata*) with a common cultivated rice (*O. stativa*) in an effort to foster rice perennialization [20,21]. In particular, perennial rice can adapt to high-altitude areas with low minimum temperatures and subhumid areas with low annual rainfall [22]. Moreover, it can be harvested for eight seasons in four years, which can be considered a promising perennial crop farming system [15,22,23]. Fujian is located in the southeast of China, with a large population and little cultivated land but rich in mountainous and photothermal resources and renowned for the cultivation of ratooning rice [24]. Via the major science and technology project of the Ministry of Science and Technology of China, an introductory experiment on perennial rice cultivars was conducted in Fuzhou, central Fujian province (26°08′ N~119°28′). Our experimental results suggest that the introduced perennial rice PR25 can safely overwinter and boost production performance and can be used as an elite variety for consecutive rice regrowing, but the regenerative property of perennial rice was not found in the annual hybrid rice AR2640 due to failure in overwinter under the perennial cropping system. In terms of yield, that of AR2640 was significantly higher than that of PR25, which can be attributed to the lower number of effective panicles and grains per panicle of PR25 than that of AR2640, reflecting the differences in genetic characteristics of the two varieties. However, the regeneration rate of PR25 was significantly higher than that of AR2640, and different from AR2640, the regeneration rate did not increase with the increase in stubble height, and the maximum regeneration rate was obtained at a lower cutting stubble height (5–10 cm) primarily due to the higher effective panicles generated from one to two nodes of the stem base in PR25 (Figure 2, Table 2, Table 3 and Table 4). The findings suggest that a reasonable stubble height of perennial rice must be fully considered.

Phytohormones (e.g., ABA, IAA, SL, etc.) can play decisive roles in governing plant growth, development, and yield by regulating various biochemical and physiological processes of crops. However, the response of phytohormones to varying stubble heights and how this response influences the regenerative characteristics of ratoon rice remain poorly documented. In this study, the grain yield of AR2640 was significantly higher than that of PR25, which could be ascribed to the marked increase in the level of vital ratoon rice hormones (ABA, IAA, and GA_3_), as these hormones are known for their potential roles in regulating crop growth, development, and yield performance [25]. The ability of AR2640 to produce a highly effective number of panicles and grains per panicle may also have contributed to this behavior [26] and corroborate previous findings [20]. These authors extensively reviewed ratoon rice farming systems and reported that long ratooning bud growth and the high survival rate subjected to panicle-removing treatment were ascribed to photosynthate allocation of the ratoon buds and the biomass of the crop stored in the stem and sheath. They further pointed out that the dry matter supplied after the main crop’s full heading played a crucial role in ratooning bud growth and development. Our findings draw parallels with the study conducted by Xu et al. [27], in which it was documented that rice varieties with 160–190 spikelets per panicle promoted high total yields in both main and ratoon crops. This behavior is consistent with Yang et al.’s findings, wherein the number of effective panicles was a crucial factor influencing rice ratooning ability [28]. 

Overwintering crops have shown great potential in promoting food security [29] and preventing the depletion of soil resources in ecosystem restoration [30]. Moreover, Lin et al. proved that the residual root system of overwintering crops was vital to promoting the growth characteristics of ratoon crops [31]. Likewise, PR25 successfully overwintered, exhibiting an average yield of 58.28% in the first ratooning season with different stubble heights primarily due to its robust root activity, as shown in Table 6. This result agrees with a previous study wherein it was established that the main crop root system played a decisive role in promoting the overwintering of ratoon rice. 

Auxin is among the fundamental phytohormones involved in modulating growth and development. Mellor et al. [25] reported that IAA oxidation via transcriptional control was crucial in *Arabidopsis thaliana* growth and development. Analogously, it was reported that an IAA-producing plant growth regulator triggered effective plant growth promotion [32]. We observed that IAA significantly increased PR25 and AR2640 on day 17, especially at a low stubble height, in agreement with Lin et al.’s findings [33]. This suggests that IAA played an indispensable role in promoting ratoon crop growth, development, and yield, as is evident in the pattern shown in Table 3, especially 17 d after harvest. This also implies that IAA was instrumental in promoting crop traits, including ratoon rice root activity, as mentioned in Table 6, especially for PR25, which conforms with previous work [34]. 

Previous studies revealed that a high GA_3_ content triggered germination, growth, the optimization of yield components and survival rate, and yield promotion of ratooning buds [35,36,37]. Here, GA_3_ peaked considerably in both varieties on day 17, especially at a low stubble height. This phenomenon played a decisive role in promoting ratoon rice traits, including PR25 yield, effective tillers, and root activity, as demonstrated in Table 2, Table 3 and Table 6, including Figure 3, where both cultivars’ regenerative tillers peaked considerably with time. The stubble height of the crop may have contributed to this behavior [20], as crop heights, which are determined by internode elongation, regulated by genes involved in brassinosteroid and GA biosynthesis or related signaling, not only determine plant resistance to crowding and lodging but also promote crop architecture, apical dominance, biomass, and mechanical harvesting [38], which is consistent with previous findings wherein the single cutting of a rice ratoon enhanced grain yield [39].

A large body of studies have reported that ABA serves many vital functions in plants, including growth regulation, stress response, and leaf senescence [40]. For instance, a recent study mentioned that a number of crucial plant hormones, including ABA and jasmonic acid, contributed to the promotion of sugarcane ratoon crop growth and soil quality [41]. In a related study, the ABA content in rice crops increased steadily, triggering a sharp increase in the crop panicle weight 20 days after heading [42]. Our investigation showed that ABA exhibited a linear increase in both PR25 and AR2640, exhibiting the highest peak 22 d after harvest, especially at low stubble heights, corroborating He et al.’s work [43], wherein they documented that ABA was responsible for ratoon rice panicle regeneration. This behavior confirms Fallah et al.’s recent findings, wherein they mentioned that ABA enrichment contributed to the promotion of sugarcane ratoon crop growth [41]. 

Over the past few decades, the diverse roles of SL in plant growth activities have been well documented [41]. On the other hand, it is still unknown how SL responds to varying stubble heights and how this response influences the performance of ratoon rice varieties. Our findings suggest that SL was responsive to ratoon rice at different stubble heights. This effect was pronounced after the last harvest, followed by the first harvest, in AR2640 at lower stubble heights. These findings align with results reported by Sun et al. [44] and Cheng et al. [45], which pointed out the crucial roles of SL in enhancing crop root architecture, hypocotyl elongation, crop growth, and seed germination. Therefore, the strong response of these essential plant hormones may play decisive roles in promoting ratoon rice trait performance, especially PR25 at low stubble heights, which agrees with previous findings [46].

## 4. Materials and Methods

### 4.1. Experimental Design and Field Management 

A randomized block trial with three replicates was conducted at the experimental farm of the College of Agriculture, Fujian Agriculture and Forestry University (FAFU), located in Fuzhou, Southeast China (26°08′ N, 119°28′ E), from 2019 to 2020. The pot experiment was conducted to observe the growth period (Table 2) of PR25 and PR107 in 2018, which was used for reference in the formal field experiment conducted in 2019–2020. The perennial rice varieties PR25 and PR107 were provided by Yunnan University, China. The seedlings were raised in a wet nursery from 22 March and transplanted into pots containing 4 seedlings on 28 April 2018. A total of 10 buckets with a diameter of 30 cm and a height of 38 cm were used. Nitrogen, phosphorus, and potassium fertilizers were applied to each tank according to the previous method [47]. 

Based on the pot experiment results, PR25 was selected as the experimental material, since the growth duration of PR107 was too long to use in the production area of the rice ratooning system. The annual indica–japonica hybrid rice AR2640, which is widely grown in AR ratooning in Fujian province, was used as a control. On the basis of the preliminary results, seedlings of the two varieties, PR25 and AR2640, were cultured in a wet nursery from 28 March 2019 and transplanted with one seedling on 28 April 2019 at a spacing of 19.8 cm × 19.8 cm based on the weather conditions, as mentioned above (Figure 1). A two-factor randomized block design experiment was conducted, beginning in March 2019 and ending in November 2020, with the two varieties of AR2640 and PR25 and three levels of stubble cut height (5 cm, 10 cm, 15 cm) in three replicates. Each plot was 4 m × 5 m, separated by 30 cm × 40 cm field dikes. The plots were covered with black plastic film and inserted into both side soils of the field diked at a depth of 20 cm to prevent nutrient loss and exchange irrigated water between plots. After PR25-FRR and AR2640-FRR were harvested in November 2019, three cutting heights of the rice stubble at 5 cm, 10 cm, and 15 cm were sequentially used for cycle regenerating in the next year (2020), although AR2640 failed to overwinter after the ratooning rice was harvested in November 2019. Overwintering evaluation of the PR and AR cultivars was performed under natural conditions without any protection other than wet soil in situ during wintertime. 

In total, both PR25-MC and AR2640-MC received 225 kg·hm^−2^ of N fertilizers in 2019, which were applied as basal, primary tillering, secondary tillering, booting, and heading fertilizers, with a splitting ratio of 4:1:1:3:1. The basic dressing superphosphate (containing 12.0% P_2_O_5_) was applied at a dosage of 562.5 kg·hm^−2^ based on the N:P:K ratio of 1:0.5:0.8. Additionally, 300 kg·hm^−2^ of potassium chloride (containing 60.0% K_2_O) was divided into the basal and booting fertilizers at a ratio of 1:1. In the ratooning season, 180 kg N·hm^−2^ of nitrogen fertilizers were used in total, of which 90 kg of N fertilizers were applied 15 d before the MC was harvested; 60 kg and 30 kg of N fertilizers were supplied 3 d and 10 d after the harvesting of the MC, respectively; and the ratio of N:P:K in the fertilization was the same as that of the MC in 2019. PR25-FRR was able to successfully overwinter under three different stubble cut heights. The fertilization pattern of the regenerating rice PR25-SRR from PR25-FRR and PR25-TRR from PR25-SRR in 2020 was the same as that of the main and ratoon rice crops in 2019. The site has a clay–loam soil with a pH of 6.18, an organic matter content of 29.5 g·kg^−1^, a total nitrogen content of 1.86 g·kg^−1^, an available phosphorus content of 42.8 mg·kg^−1^, and an available potassium content of 90.1 mg·kg^−1^. 

Irrigation was performed according to the method described in a previous study [48]. To prevent yield loss, pesticides and insecticides were applied two or three times during the first season and once during the replanting season to control pests and diseases. The weather information of the site, including daily minimum and maximum temperatures and precipitation, was collected from a weather station located near the experimental field during the rice growth season.

### 4.2. Determination Items and Methods

#### 4.2.1. Observation of Growing Stage and Tillering Dynamics

From the beginning of rice sowing, the main growth stages of rice were recorded, including the sowing, transplanting, tillering, heading, and ripening stages of the MC in 2019. After the transplanting and recovery of the rice seedlings, the tillering dynamics and main growth stages of 10 plants in each treatment were randomly selected for us to observe and record. After the harvest of the main crop, the regenerative rates from the axillary buds in the remaining stubble were assessed at an interval of 5 d. The regenerative rates were calculated using the following formula:

Regeneration rate (%) = (number of effective tillers in ratooning season rice/number of stubble stems after harvest of the main crop) × 100

#### 4.2.2. Dynamic Observation on Sprout and Elongation of Axillary Buds

Zero, ten, and twenty days after the MC harvest, three plants were randomly sampled from each plot and the stems were dissected to record the sprout and elongation of the axillary buds. The sprout and elongation of the axillary buds were also evaluated 2 d, 7 d, 12 d, 17 d, and 22 d after the harvest.

#### 4.2.3. Determination of Root Activity

During the tillering, booting, full heading, and ripening stages of the main crop and at the heading and ripening stages of the first ratooning season, three rice plants with uniform growing and tillering statuses were randomly selected from each plot. The rice roots were dug from the remaining stubble base of the rice as the center and then put into a net bag and washed with water and cleaned with distilled water again. Subsequently, the water on the root surface was sucked up with tissues for immediate testing using the alpha-naphthalamine method.

#### 4.2.4. Determination of Endogenous Phytohormones 

On days 2, 7, 12, 17, and 22 after harvesting the main crop, the axillary buds were sampled from the first to the fourth nodes. Three rice plants were selected to sample the axillary buds at the high node positions (3rd and 4th) and the low node positions (1st and 2nd), and the axillary buds at the same node were mixed, wrapped in tin foil, labeled, and immediately frozen in liquid nitrogen. They were then taken back to the laboratory and stored in a refrigerator at −80 °C for the purpose of determining endogenous phytohormones.

Phytohormones were detected following the method described by Huang et al. (2020) [49]. In brief, the fresh samples of axillary buds sampled from higher- and lower-position nodes were placed in a pre-cooled mortar and ground to powder with liquid nitrogen, respectively. Then, 0.1 g of the powder was collected and transferred to three centrifugal tubes (it is worth noting that we tried to keep a small amount of liquid nitrogen in the mortar to prevent hormone degradation during the weighing process). Subsequently, 750 µL of 80% pre-cooled methanol (chromatographic grade) were added, soaked, and placed in a 4 ℃ shaker for 10 h. All samples were centrifuged at 5000 rpm·min^−1^ at 4 °C for 15 min. The supernatant was transferred to a new tube, and this extraction was repeated twice. Finally, all of the supernatants were mixed and concentrated to 100–200 µL using a rotary evaporator at 35 °C and then brought to a volume of 250 µL with chromatographic methanol. After that, the homogenized extraction was passed through a 0.45 µm filter membrane and then stored at −20 °C or −80 °C for further detection via high-performance liquid chromatography (HPLC). Grinding and extraction were performed away from light to avoid hormone degradation.

Using CBM-20 A, LC-20 A, and SPD-20 A (Shimadzu, Kyoto, Japan) with C18 columns (4.6 mm × 250 mm, ID 5 µm, INERTSUSTAIN), high-performance liquid chromatography was performed, with the parameters set as follows: column box temperature, 35 °C; detection wavelength, 254 nm; flow velocity, 1.0 mL/min and 0.8 mL/min; injection volume, 20 µL. The external standard method for quantitative analysis was adopted. HPLC elution procedure setting: mobile phase, A—acetonitrile (chromatographic pure); B—0.6% glacial acetic acid (chromatographic pure).

#### 4.2.5. Determination of Rice Yield and Its Components

For the determination of grain yield, five representative rice plants were randomly sampled in each experimental plot at the maturity stage of the two rice varieties, including the main and ratoon rice crops, in short-cycle regeneration cultivation in 2019–2020. The number of effective panicles, the number of grains per panicle, the seed-setting percentage, the 1000-grain weight, and the actual yield of the plot were measured for each treatment. 

#### 4.2.6. Data Processing and Analysis

Microsoft Excel 2016 was used for preliminary data processing. IBM SPSS Statistics 22.0 was employed for statistical analysis using the LSD method to test the level of significant difference between blocks, and Microsoft Excel 2016, Origin 8.0, and GraphPad prism 9.0 were used for mapping the data processing.

## 5. Conclusions

Our findings provide evidence of how ratoon rice, subjected to different stubble heights, displays enhanced performance. This enhancement is primarily attributed to the strong response of plant growth hormones, including ABA, IAA, SL, and GA_3_, to the lower stubble heights of ratoon rice. This study provides valuable insights and crucial references for delving deeper into cultivating high-yielding perennial rice. Further studies using different crops conducted over a protracted period are warranted to validate these results. 

## Figures and Tables

**Figure 1 plants-12-03910-f001:**
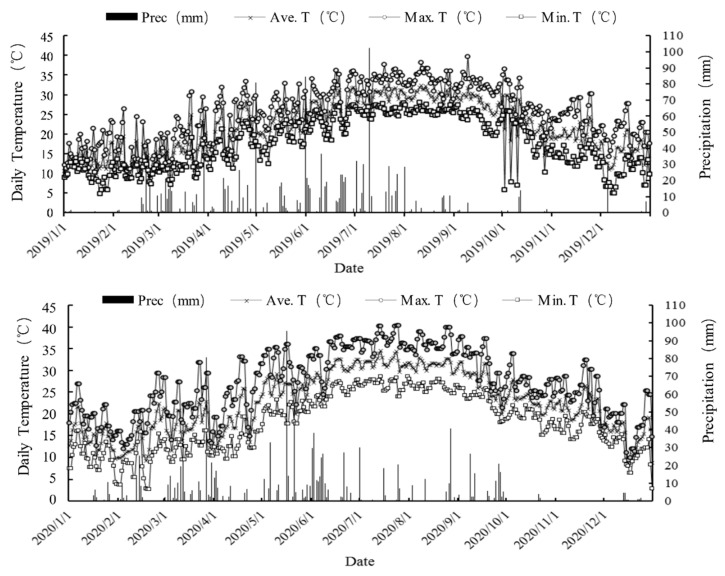
The daily changes in annual main climatic factors monitored in situ during the entire duration of rice growth from 2019 to 2020.

**Figure 2 plants-12-03910-f002:**
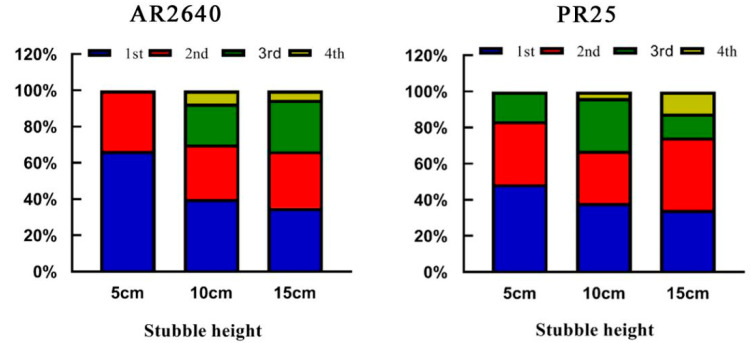
Yield contribution rate of panicles from different nodes of the main crop to first ratooning season rice.

**Figure 3 plants-12-03910-f003:**
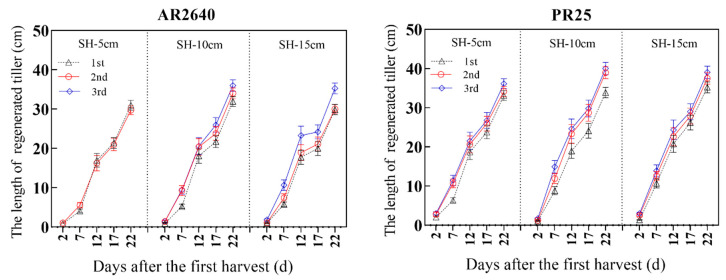
The length of regenerated tillers at different nodes after the first harvest under different treatments. Note: 1st, 2nd, and 3rd refer to axillary buds of the first, second, and third node from the base, respectively; SH-5 cm, SH-10 cm, and SH-15 cm refer to the stubble heights of 5 cm, 10 cm, and 15 cm after harvesting the main crop, respectively.

**Figure 4 plants-12-03910-f004:**
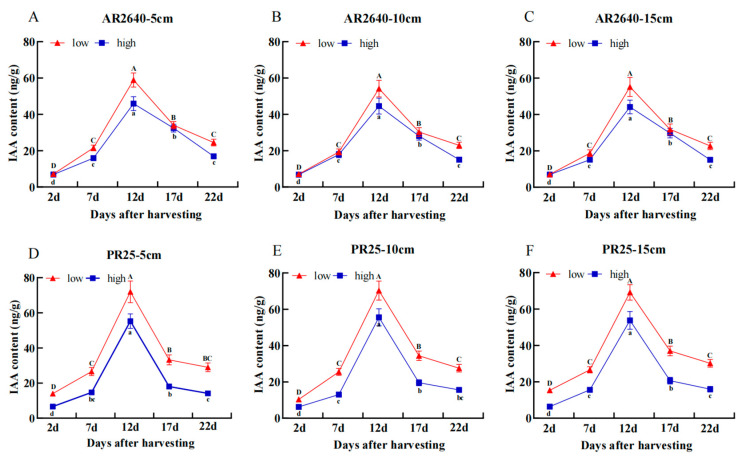
Dynamic changes in average IAA content in regenerated buds of each node in AR2640 and PR25 with different stubble height treatments. AR2640 were subjected to 5 cm (**A**), 10 cm (**B**), 15 cm (**C**) and PR25 were subjected to 5 cm (**D**), 10 cm (**E**), 15 cm (**F**), “low” stands for low axillary buds, “high” stands for high axillary buds.

**Figure 5 plants-12-03910-f005:**
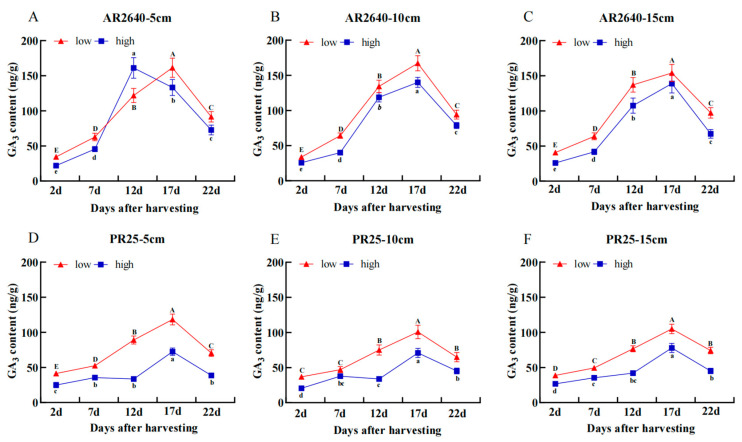
Dynamic changes in GA_3_ content of axillary buds at different nodes in AR 2640 and PR 25 with different stubble heights. AR2640 were subjected to 5 cm (**A**), 10 cm (**B**), 15 cm (**C**) and PR25 were subjected to 5 cm (**D**), 10 cm (**E**), 15 cm (**F**), “low” stands for low axillary buds, “high” stands for high axillary buds.

**Figure 6 plants-12-03910-f006:**
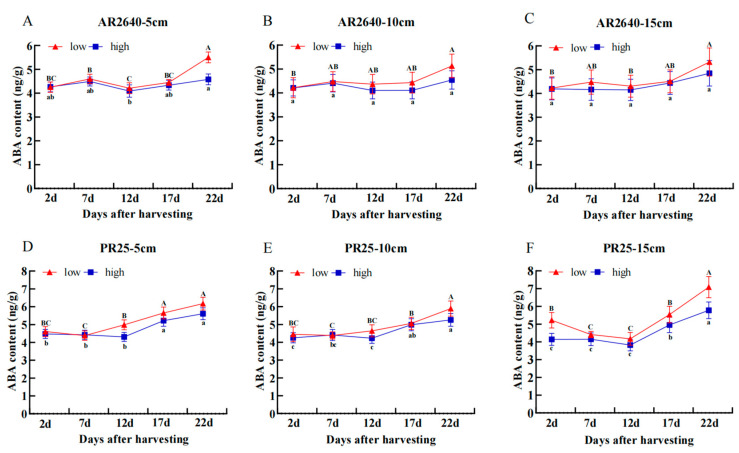
Dynamic changes in ABA content of regenerated buds at different nodes in AR 2640 and PR25 at different stubble heights. AR2640 were subjected to 5 cm (**A**), 10 cm (**B**), 15 cm (**C**) and PR25 were subjected to 5 cm (**D**), 10 cm (**E**), 15 cm (**F**), “low” stands for low axillary buds, “high” stands for high axillary buds.

**Figure 7 plants-12-03910-f007:**
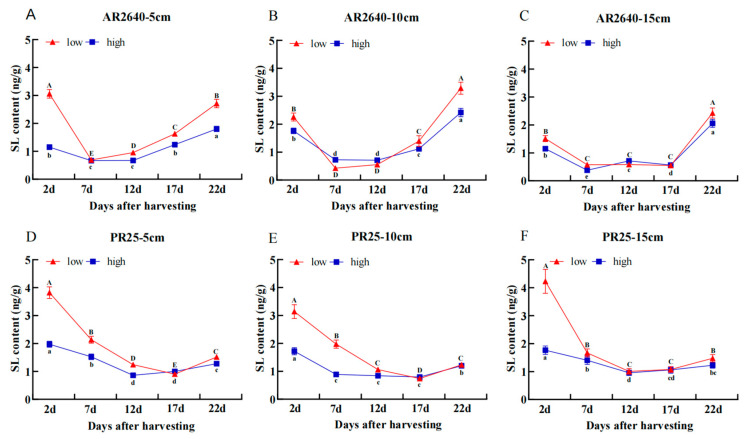
Dynamic changes in SL content of regenerated buds at different nodes in AR2640 and PR25 at different stubble heights. AR2640 were subjected to 5 cm (**A**), 10 cm (**B**), 15 cm (**C**) and PR25 were subjected to 5 cm (**D**), 10 cm (**E**), 15 cm (**F**), “low” stands for low axillary buds, “high” stands for high axillary buds.

**Table 1 plants-12-03910-t001:** Experimental trial design and basic information.

Cropping System (Year)	Variety	Stubble Height (cm)	Transplanting/Sprouting (Day/Month)	Fully Heading (Day/Month)	Ripening (Day/Month)	Growth Duration (Day)
MC (2019)	AR2640	/	28/4	14/7	18/8	147
MC (2019)	PR25	/	28/4	14/7	22/8	151
FRR (2019)	AR2640	5	26/8	3/10	6/11	80
		10	26/8	1/10	2/11	76
		15	26/8	28/9	30/10	73
FRR (2019)	PR25	5	28/8	8/10	13/11	83
		10	28/8	7/10	12/11	82
		15	28/8	6/10	11/11	81
SRR (2020)	PR25	5	2/4	27/6	6/8	126
		10	2/4	26/6	6/8	126
		15	2/4	26/6	6/8	126
TRR (2020)	PR25	5	12/8	30/9	8/11	82
		10	12/8	30/9	6/11	80
		15	12/8	30/9	6/11	80

Note: MC: main crop; FRR: first regenerative rice; SRR: second regenerative rice; TRR: third regenerative rice; PR: perennial rice; AR: annual hybrid rice.

**Table 2 plants-12-03910-t002:** Yield and its composition of different rice accessions.

Year	Ratooning System	Cultivar	Stubble Height (cm)	Effective Panicles (10^4^·hm^−2^)	Grains Panicle^−1^	1000-Grain Weight (g)	Seed Setting Percentage (%)	Grain Yield (t·hm^−2^)
2019	MC	AR2640	/	200.1 d	240.0 a	22.1 d	82.6 a	8.77 b
		PR25	/	185.6 e	180.0 b	25.2 a	86.5 a	7.28 c
	FRR	AR2640	5	205.1 d	88.5 i	21.8 d	62.9 d	2.49 g
			10	225.1 c	86.1 i	21.9 d	70.8 c	3.01 f
			15	270.1 b	83.5 ij	22.1 d	78.9 b	3.93 e
		PR25	5	265.1 b	110.3 f	24.6 b	79.0 b	5.68 d
			10	260.1 b	126.5 e	24.2 b	75.0 b	5.97 d
			15	230.1 c	97.3 gh	24.3 b	60.0 e	3.26 f
2020	SRR	AR2640	5	0.0	0.0	0.0	0.0	0.0
			10	0.0	0.0	0.0	0.0	0.0
			15	0.0	0.0	0.0	0.0	0.0
		PR25	5	331.6 a	167.9 c	23.4 c	82.1 a	10.70 a
			10	307.4 a	170.5 c	23.8 c	82.4 a	10.28 a
			15	290.6 b	154.3 cd	23.3 c	83.4 a	8.71 b
	TRR	PR25	5	185.6 e	112.3 f	24.5 b	76.5 b	3.91 e
			10	192.6 e	120.5 e	23.5 c	73.8 b	4.15 e
			15	185.6 e	102.6 g	24.5 b	67.8 d	3.15 f
			df					
	F-value		15	71.4 ***	88.9 ***	60.8 ***	63.1 ***	99.4 ***

Note: *** represents significant levels of *p* < 0.001. MC: main crop; FRR: first regenerative rice; SRR: second regenerative rice; TRR: third regenerative rice; PR: perennial rice; AR: annual hybrid rice. The data with different lowercase letters in the same column indicate significant differences at the 5% probability level.

**Table 3 plants-12-03910-t003:** The effect of different stubble heights on the percentage of effective tillers of AR2640 and PR25 in a short-cycle ratooning system.

Year	Cultivar	Ratoon Crop System	SH	MS	HT	EP	RR	PEP
			(cm)	(10^4^·hm^−2^)	(10^4^·hm^−2^)	(10^4^·hm^−2^)	(%)	(%)
2019	AR2640	FRR	5	200.1 d	282.6 d	205.1 e	102.5 e	72.6 c
			10	200.1 d	315.2 c	225.1 e	112.5 d	71.4 c
			15	200.1 d	337.7 c	270.1 d	135.0 b	80.0 a
	PR25	FRR	5	185.6 e	332.6 c	265.1 d	142.8 a	79.7 a
			10	185.6 e	326.6 c	260.1 d	140.1 a	79.6 a
			15	185.6 e	300.2 d	230.1 e	124.0 c	76.6 b
2020	PR25	SRR	5	265.1 b	432.7 a	331.6 a	125.1 c	76.6 b
			10	260.1 b	431.5 a	307.4 b	118.2 d	74.2 b
			15	230.1 c	387.2 b	290.6 c	126.3 d	76.6 b
	PR25	TRR	5	331.6 a	264.8 e	185.6 f	55.97 f	70.1 c
			10	325.2 a	274.1 e	192.5 f	59.19 f	70.5 c
			15	290.6 a	260.7 e	185.6f	62.79 f	71.2 c

Note: MC: main crop; FRR: first regenerative rice; SRR: second regenerative rice; TRR: third regenerative rice; PR: perennial rice; AR: annual hybrid rice; SH: stubble height; MS: the number of mother stems; HT: the highest tillers; EP: the number of effective panicles; RR: the regeneration rate; PEP: the percentage of effective panicles. The data with different lowercase letters in the same column indicate significant differences at the 5% probability level according to the LSD test.

**Table 4 plants-12-03910-t004:** Changes in the length of axillary buds regenerated from the booting stage to the mature stage of the AR2640 and PR25 main crops.

Cultivar	Basal Node	BS	HS	RS	
		BL (mm)	BL (mm)	BL (mm)	INL (cm)
	3rd	8.08 a	14.33 a	22.24 a	7.80 a
AR2640	2nd	5.52 b	6.90 c	15.63 b	5.23 b
	1st	6.28 b	9.88 b	21.69 a	1.40 d
	3rd	6.80 b	13.25 a	12.75 c	6.04 b
PR25	2nd	8.87 a	6.43 c	12.53 c	4.05 c
	1st	5.89 b	7.90 c	14.79 b	1.15 d

Note: BS: booting stage; HS: heading stage; RS: ripening stage; BL: bud length; INL: internode length. The data with different lowercase letters in the same column indicate significant differences at the 5% probability level according to the LSD test.

**Table 5 plants-12-03910-t005:** Correlation coefficient between the endogenous hormone content and regeneration rate of regenerated buds after harvesting the main crop.

	R_2d_	R_7d_	R_12d_	R_17d_	R_22d_
GA	−0.413 *	−0.230	−0.350	0.580 *	−0.263
IAA	−0.434 *	0.091	−0.325	0.202	−0.363
ABA	−0.490 *	−0.029	−0.047	−0.318	0.083
SL	−0.617 *	0.516 *	−0.125	0.677 *	0.046
IAA/ABA	−0.404	0.101	−0.384	0.346	−0.517 *
(IAA + GA)/ABA	−0.364	−0.151	−0.435 *	0.550 *	−0.321
IAA + GA	−0.451 *	−0.148	−0.446 *	0.549 *	−0.292
GA/ABA	−0.301	−0.241	−0.338	0.568 *	−0.319

Note: * represents significant levels of *p* < 0.05. The asterisk denotes a significant correlation at the *p* = 0.05 significance level. R_2d_, R_7d_, R_12d_, R_17d_, and R_22d_ refer to the regenerative rate 2 d, 7 d, 12 d, 17 d, and 22 d after harvesting the main crop, respectively.

**Table 6 plants-12-03910-t006:** Root activity in the first and second season (µg·g^−1^·h^−1^·FW).

Cultivar	MBS	MHS	MMS	SH (cm)	RBS	RHS	RMS
AR2640	9.94 b	5.83 b	4.53 b	5	10.00 b	8.82 b	7.35 b
				10	9.98 b	8.63 b	7.12 b
				15	10.2 b	8.98 b	7.41 b
PR25	12.4 a	7.34 a	5.91 a	5	12.86 a	11.62 a	10.03 a
				10	12.83 a	11.58 a	9.97 a
				15	12.78 a	11.42 a	9.72 a

Note: MBS: the booting stage of the main crop; MHS: r the heading stage of the main crop; MMS: the milky stage of the main crop; RBS: the booting stage of the first ratooning season; RHS: the heading stage of the first ratooning season; RMS: the milky stage of the first ratooning season. The data with different lowercase letters in the same cultivar under the same treatment indicate a significant difference at the 5% probability level.

**Table 7 plants-12-03910-t007:** Correlation analysis between the grain yield of the second rice harvest, the regeneration rate, and the root activity.

Item	R-MBS	R-MHS	R-MRS	R-RBS	R-RHS	R-RRS	RR
RR	0.952 **	0.887 **	0.935 **	0.861 *	0.869 **	0.971 **	1.000
RY	0.857 *	0739 *	0.846 **	0.700 *	0.546 *	0.695 *	0.921 **

Note: *, ** represent significant levels of *p* < 0.05 and 0.01, respectively. R-MBS: the root activity in the main crop at the booting stage; R-MHS: the root activity in the main crop at the heading stage; R-MRS: the root activity in the main crop at the maturity stage; R-RBS: the root activity in the first ratooning season at the booting stage; R-RHS: the root activity in the first ratooning season at the heading stage; R-RRS: the root activity in the first ratooning season at the maturity stage; RR: the regenerative rate; RY: the yield of ratooning rice.

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
