# Peer review of "Physiological Properties of Perennial Rice Regenerating Cultivation in Two Years with Four Harvests"

_plants, 2023, doi:10.3390/plants12223910_

Round 1

Reviewer 1 Report

Comments and Suggestions for Authors

In summary, this study addresses an important research gap in the context of crop perennialization and provides valuable insights into the regenerative characteristics of perennial rice. It would benefit from some improvements in terms of clarity and conciseness, as well as a more explicit statement about the practical implications of the findings. Some minor issues should be addressed before publication.

Length: The abstract could be more concise. Some sentences are long and complex, which may make it less reader-friendly. It's important to convey the key points succinctly.

Clarity: While the abstract provides an overview of the study, some sentences, especially in the latter part, could be rephrased for clarity. For example, the sentence "Further analysis revealed that PR25 exhibited a higher regenerating rate from the lower position nodes..." could be simplified for better understanding.

Recommendations: The abstract doesn't explicitly state the implications of the findings or any recommendations. Adding a sentence about the practical implications of the research would enhance its impact.

Introduction

Check the sentences. Rapid global population expansion has made finding more varied food and energy sources to support human well-being even more important.

Line 35-37. Also because of future climate change. The demand for food such as rice is anticipated to rise dramatically as the world population is projected to reach 9 bil-lion people by 2050[1] and the climate change [2].

Could cite [2]c Liu, K., Harrison, M.T., Yan, H. et al. Silver lining to a climate crisis in multiple prospects for alleviating crop waterlogging under future climates. Nat Commun 14, 765 (2023). https://doi.org/10.1038/s41467-023-36129-4

Please describe the objectives clearly at the end of introduction.

Fig. 1. Check the rainfall in the upper panel. Daily rainfall was greater than 100mm?

Comments on the Quality of English Language

Need to improve

Reviewer 2 Report

Comments and Suggestions for Authors

Comments and Suggestions for Authors 

The paper ‘Physiological properties of perennial rice regenerating cultivation in two years with four harvests’ is an interesting study that explore the regenerative characteristics and physiological mechanisms of an annual hybrid rice (AR2640) and a perennial rice (PR25) subjected to different stubble heights (5, 10, and 15 cm).

This work provides novel insights into the develop breeding programs and provide valuable insights for delving deeper into cultivating high-yielding perennial rice.

This work is interesting and useful, however, there are the following comment and suggestions that may be addressed before the MS is accepted for publication.

In the keywords I suggest to insert ‘ratoon crop’.

Table 1.: please report the captions in a more concise way, for example 'MC: main crop; FRR: first regenerative rice; SRR:....etc', where possible it would be useful directly in the title of the table itself and please also repeat the meaning of PR and AR in the captions.

Table 2.: statistical analysis relating to column ‘Grains panicle’ seems a bit strange (180c and 167.9b), please check it or explain. Moreover, I recommend formatting the table better, that is, reporting the parameters in the same way as the others Tables by removing the vertical lines, and if possible insert horizontal lines that at least separate the years, it should make understanding a little clearer.

Table 3.: please report the captions in a more concise way, as request for Tab.1 and insert horizontal lines to  separate the years.

Figure 3: it would be useful to differentiate the nodes with gray scales or different pattern fills instead of colors

Table 4.: statistical analysis relating to column ‘HS’ seems a bit strange (6.90c and 6.43b, ‘in the same column’), please check it or explain.

Paragraph 2.3.3.: I suggest not starting sentences with ‘Figure’ and reformulating the sentences and I don't think it's necessary to include sub-paragraphs for each phytohormone.  Furthermore, throughout the paragraph it is often reported that the differences between the various concentrations of phytohormones are significant or not (for example Line 225, 229,etc.) it would be useful to include these significances in the figures. At the same time, if possible, please increase the size of figures 4,5,6 and 7.

Line 277: regarding table 5 on correlations, it would be interesting to know if and how the perennial cultivar differs from the annual one.

Tables 6 and 7: please report the captions in a more concise way, as request for Tab.1

Line 346: what previous works? please indicate some literature about perennial rice or perennial cereals in general.

Line 510: please write the conclusion in capital letters.

In the paragraph relating to the Discussion I didn’t find many references regarding perennial rice, but mainly about ratoon rice in general, even though the title of the article itself is about perennial rice. Furthermore, it would be useful to add a sentence that points out what differences, if any, have been found between the perennial and annual plants. Same comment for the Conclusions paragraph, which doesn’t highlight any peculiarities of perennial rice compared to annual rice as it was reported in Abstract section.

Author Response

Thanks for your comments and suggestions for our manuscript summitted to your “Plants” journal. Based on the comments, we have corrected and improved it as shown in our revised paper. We wish the revised paper can be qualified to requirement for final publication. Here, the revised explanation was listed as the following responses to the reviewer’s comments in one by one. 

Reviewer 3 Report

Comments and Suggestions for Authors

The publication discusses the recent global interest in crop perennialization for sustainable agriculture. It highlights the lack of detailed information regarding the regenerative characteristics and physiological mechanisms of perennial rice in crop ratooning systems with different rice stubble heights. Overall, findings from the reviewed article provide valuable insights for the cultivation of high-yielding perennial rice and underscore the importance of stubble height and plant hormone responses in achieving sustainable crop perennialization.

Below please find some remarks to your text:

Tables 2 and 3 must fit to one page each. It is unacceptable to split the table across two pages.

The sentence from row # 399 – 400 (“Yunnan University, China, provided the perennial rice variety varieties PR25 and PR107”) was repeated in row 403 (“The perennial rice variety PR25 and PR107 provided by Yunnan University, China”).

Author Response

(The authors gave the same response as above.)
